# Integrated Analysis of the miRNA-mRNA Regulatory Network Involved in HIV-Associated Neurocognitive Disorder

**DOI:** 10.3390/pathogens11040407

**Published:** 2022-03-27

**Authors:** Chang Liu, Qian Ding, Xiaohong Kong

**Affiliations:** School of Medicine, Nankai University, No. 94, Weijin Road, Nankai District, Tianjin 300071, China; 15646735691@163.com

**Keywords:** miRNA, mRNA, HIV, network, bioinformatics, HAND, viral infection, CNS damage

## Abstract

HIV-associated neurocognitive disorder (HAND) is an array of neurocognitive changes associated with HIV infection, and the roles of microRNAs in HAND have not yet been completely revealed. Based on published data and publicly available databases, we constructed an integrated miRNA-mRNA network involved in HAND. Bioinformatics analyses, including gene ontology, network analysis, and KEGG pathway analysis, were applied for further study of the network and the genes of the network. The axon guidance KEGG pathway, three genes *NTNG1*, *EFNB2*, *CXCL12*, and 17 miRNAs which regulate these genes are spotlighted in our study. This study provides new perspectives to the knowledge of miRNAs’ roles in the progression of HAND, and our findings provide potential therapeutic targets and clues of HAND.

## 1. Introduction

Although the HIV/AIDS epidemic remains a significant global health challenge [1], antiretroviral therapy (ART) has transformed HIV infection from a fatal disease into a chronic illness [2]. ART achieves HIV suppression, but HIV-associated neurocognitive disorder (HAND) has been one of the most prevalent comorbidities in the era of ART [3]. HAND comprises a spectrum of conditions ranging from asymptomatic neurocognitive impairment to mild neurocognitive disorder to HIV-associated dementia (HAD), the clinical correlate of HIV encephalitis [4,5]. During HIV infection, neuronal damage is likely a bystander effect of infected cells since neurons are not HIV-permissive cells [6]. HIV-infected macrophages and lymphocytes can cross the blood–brain barrier and infect glial cells [7], so HIV infection is restricted to microglia and astrocytes in the central nervous system (CNS) where resident lymphoid cells are absent [8]. In the CNS, HIV-infected cells release a plethora of host and viral molecules including chemokines, cytokines, and viral proteins; these molecules promote neuroinflammation even during ART while HIV replication is undetectable [9]. Furthermore, some studies have reported that shed or secreted HIV proteins, such as gp120, Tat, and Vpr, participate in the neuropathogenesis of HIV [10,11,12,13].

Because neurons are not productively infected by HIV, neuronal damage is mainly through indirect mechanisms [14,15]. CNS neurons are vulnerable to damage by HIV viral proteins, and the vulnerability is largely mediated through the presence of several neuronal cellular receptors, such as chemokine receptors (CCR5 or CXCR4), the low-density lipoprotein receptor related protein (LRP), the N-methyl-D-Aspartate receptors (NMDAR), and the dopamine transporter [16,17,18,19]. HIV envelope protein gp120 incites the activation of chemokine receptors on neurons and triggers the elevation of intracellular Ca^2+^ leading to apoptosis [20]. The HIV transcriptional activator protein Tat can also bind LRP in neurons, causing LRP internalization and inducing apoptosis [21,22]; recently it has been reported that Tat exerts its neurotoxic effects by downregulating Sonic hedgehog signaling [23]. Furthermore, because ART does not impact levels of HIV Tat and the CNS is not accessible to the ART regiments, Tat has been regarded as a mediator of HAND [24]. However, the detailed molecular mechanism of how HIV damages neurons needs further exploration. Recently, accumulating evidence has suggested that microRNAs (miRNAs) also play important roles in this process [25,26,27,28].

MiRNAs are a class of endogenous, small, non-coding RNAs of 18–26 bp and they can regulate post-transcriptional protein expression via interaction with 3′UTR region of the target genes’ message RNAs (mRNAs) [29,30]. The CNS is the most complex organ of mammals and houses a remarkable diversity of miRNAs [31]; therefore, the CNS is an interesting study target for miRNA studies [32]. The CNS miRNAs play an important role in regulating gene expression during neuronal development, from neurogenesis to synaptogenesis, as well as in maintaining normal neuron function [33,34,35]. Moreover, several reports have suggested that miRNAs are involved in HIV-related neural injury: Rahimian et al. reported that HIV Tat significantly induces the expression of miR-132, downregulates target genes of miR-132 in astrocytes and neurons, and causes neurite shortening [36]; Bagashev et al. found that Tat deregulates the p73 pathway and causes neuronal dysfunction through miR-196a [37]; miR-142 is increased in neurons and myeloid cells in the brain of HAND patients, and indirectly downregulates MAOA to change dopaminergic neurotransmission [38]; miR-21 is significantly upregulated in the brains of simian immunodeficiency virus encephalitis (SIVE) monkey models, and it specifically targets MEF2C to induce neuronal dysfunction and neurodegeneration [39]. The abovementioned studies indicate that HIV can manipulate the expression of neuronal genes, via modulating the expression of particular microRNAs, and miRNAs may serve as key elements in gene regulatory networks in HAND.

In our study, an integrated miRNA-mRNA regulatory network was constructed and studied. We collated the miRNAs whose expressions have been reported to change significantly in HAND [26,27,36,37,38,39,40,41,42,43,44,45]. These miRNAs were divided into two different groups: the upregulated group and the downregulated group. The potential miRNA-target genes were predicted by three miRNA-target prediction public databases [46,47,48]. The union of each miRNA’s potential miRNA-target genes in different database was constructed, so a mini miRNA-target genes of HAND database was established. In order to construct a miRNA-mRNA regulatory network involved in HAND, a Gene Expression Omnibus (GEO, https://www.ncbi.nlm.nih.gov/geo/ (accessed on 1 December 2021)) dataset was reanalyzed to obtain the differential expression genes (DEGs) in neurons affected by HIV. These DEGs were used as a screening condition to construct the miRNA-mRNA network; then, the network was analyzed and demonstrated by Cytoscape [49] and R packages [50]. These miRNA-target genes in the network were performed to construct a protein–protein interaction (PPI) network, and the genes of the PPI network were applied KEGG pathway enrichment. The axon guidance pathway (hsa04360) was enriched, the miRNAs and mRNAs/genes of our network participating in this pathway were found. Our study generated a holistic picture of regulatory miRNA-mRNA involved in HAND, and some potential important miRNAs and mRNAs/genes of the network could be regarded as therapeutic targets or agents for the treatment of HAND in the future.

## 2. Results

An overview of the research strategy is shown inz Figure 1. Briefly, we identified the differentially expressed genes (DEGs) associated with HAND based on the public GEO database, and the functional analyses were performed on DEGs. In parallel, we summarized the miRNAs associated with HAND based on the published literature, and the target genes of these miRNAs were predicted on public miRNA databases. Based on the above information, the integrated miRNA-mRNA regulatory network was constructed. By analyzings the network, we provided the potential key miRNAs, mRNAs/genes, and the potential key KEGG pathway involved in HAND. As shown in Figure 1, the locations of our main results were also indicated in advance for readers’ convenience.

### 2.1. Identification and Functional Analysis of DEGs Associated with HAND

In order to identify the significant DEGs associated with HAND, GSM1045805, GSM1045806, GSM1045807, and GSM1045808 in GSE44265 were reanalyzed by R software. Briefly, GSM1045807 and GSM1045808 are differentiated SH-5Y cells treated with supernatant from HIV infected U-937; GSM1045805 and GSM1045806 are normal differential SH-5Y cells as the negative control group. As shown in Figure 2A, a volcano plot showed the genes’ expression changing in differentiated SH-5Y cells after treatment with supernatant from HIV infected U-937; the red dots indicate the significantly changing genes with more than a 2-fold change and an adjusted *p* value of less than 0.05; and the drastically changing DEGs with more than 5.657-fold change (or 2.5 log2 fold changing) are annotated by gene name. We call these drastically changing mRNAs/genes representative DEGs.

For more detailed information as shown in Figure 2B, there are 12 representative upregulated expression mRNAs/genes and 12 representative downregulated expression mRNAs/genes. The cluster heatmap of these representative changing DEGs is presented in Figure 2B. To further understand the molecular functions of DEGs, all 296 significant DEGs were separated into two groups: the upregulated/activated mRNA-gene group (142 genes, the activated group) and the downregulated/suppressed mRNA-gene group (154 genes, the suppressed group). These genes of the two groups were taken for GO molecular function (GO-MF) enrichment analysis. As showcased in Figure 2C, some cellular channel activities, such as cation channel, ion channel, substrate-specific channel, ion gated channel, and gated channel, were significantly enriched in the suppressed group. The suppression of these channels’ activity obviously affects the functions of neurons in CNS.

### 2.2. Construction of the miRNA-mRNA Net Involved in HAND

According to the published literature about miRNA and HIV-associated neural damage, 101 upregulated miRNAs and 91 downregulated miRNAs were selected and their miRNA target mRNAs/genes were searched and predicted in three public web databases (miRtarbase database [46], miRDB database [47], and Targetscan database [48]). The upregulated-miRNA-downregulated-mRNA pairs were filtered with previously constructed downregulated/suppressed mRNAs/gene group (154 genes); and downregulated-miRNA-mRNA pairs were filtered with the upregulated/activated mRNA/gene group (142 genes) correspondingly. Finally, an upregulated-miRNA-downregulated-mRNA network, which contains 51 miRNAs and 136 mRNAs/genes and 1458 regulation connections, was constructed. Meanwhile, a downregulated-miRNA-upregulated-mRNA network, which contains 88 miRNAs and 129 mRNAs/genes and 1981 regulation connections, was also constructed. These two complete miRNA-mRNA regulation pair tables are provided as Appendix A.

To show the features of the miRNA-mRNA networks intuitively, we constructed two mini networks filtered with representative DEGs as mentioned above. As shown in Figure 3, upregulated miRNAs are indicated as red diamonds; downregulated mRNAs/genes are indicated as green dots; downregulated miRNAs are indicated as green diamonds; upregulated mRNAs/genes are indicated as red dots; pairs are connected by strain lines. The connectional characteristics of a certain miRNA or mRNA/gene in the network were significantly different. Some miRNAs, such as has-miR-527 or mir-615-5p only regulated one certain target mRNA/gene in the representative network, while some miRNAs, such as mir-7-3p or miR-1250-3p, regulated more than 5 target mRNAs/genes; the target mRNAs/genes behaved similarly. These miRNAs or mRNAs/genes, which had more connections with other genes or miRNAs, may play a more important role in the network comparing with other miRNAs or genes. We called them key miRNAs or mRNA/genes in the miRNA-mRNA network. In the network analysis, the degree of a node, which is the number of its neighbors, shows this characteristic; a node with high degree has more edges/connection than a low degree node. The node degree analysis was carried on the whole miRNA-gene network; finally, top 10 upregulated miRNAs, downregulated mRNAs/genes, downregulated miRNAs, and upregulated mRNAs/genes are listed in Table 1.

### 2.3. Functional Exploration of the miRNA-mRNA Net Involved in HAND

In order to explore the potential biological function of the genes in the miRNA-mRNA net, 136 downregulated mRNAs/genes and 129 upregulated mRNAs/genes were analyzed by ClueGo plugin of Cytoscape. As shown in Figure 4A, a protein–protein interaction (PPI) network including 34 downregulated mRNAs/genes, 18 upregulated mRNA/genes, and 8 enriched genes was constructed. The relations between these genes were activation or inhibition which is indicated as green arrows or red arrows, respectively. Based on the results of the PPI net, all 60 mRNAs/genes (34 downregulated, 18 upregulated, and 8 enriched) were used to find the key KEGG pathways. Several important signaling pathways were significantly enriched, including PI3K-Akt signaling pathway (hsa04151), Wnt signaling pathway (hsa04310), MAPK signaling pathway (hsa04010), TGF-beta signaling pathway (hsa04350), NF-kappa B signaling pathway (hsa04064). These results implicated that these mRNAs/genes participate in multiple cellular signaling pathways. 

Furthermore, the axon guidance pathway (hsa04360) was also significantly enriched; there were seven genes (three mRNAs/genes in the miRNA-mRNA network and four enriched genes in the PPI network) participating in this pathway. All seven genes were marked in red boxes in Figure 4B. Three mRNAs/genes in the miRNA-mRNA network were *NTNG1*, *EFNB2*, and *CXCL12*, which were all downregulated in the miRNA-mRNA pairs network. For clarity, all three genes are written in green with down arrows indicating downregulation. We also queried the miRNAs for these three genes with our miRNA-mRNA network, those miRNAs which regulate these three genes were shown in a Venn diagram of Figure 4C. There were 5 miRNAs regulating *EFNB2*, 5 miRNAs regulating *CXCL12*, and 12 miRNAs regulating *NTNG1*. A single miRNA may regulate multiple mRNAs/genes; for example, mir-7-3p can regulate *NTNG1*, *EFNB2*, and *CXCL12*, as indicated in Figure 4C.

## 3. Discussion

In our study, we constructed and analyzed an integrated miRNA-mRNA regulatory network involved in HAND. The miRNA information was acquired from 12 published papers about the expression changing of miRNAs during HIV-associated neuron damage. Of these studies, some were focused on some specific miRNAs and some were general miRNA screening; some were clinical case studies and some were mechanism studies at the cellular level or animal models. We collected the information of these miRNAs expecting to explore undiscovered molecular mechanisms. The mRNAs/genes information came from reanalyzing or reusing the GSE44265 GEO dataset. Based on our experimental design requirements, finally, four samples (GSM1045805, GSM1045806, GSM1045807, GSM1045808) were chosen. GSM1045807 and GSM1045808 are differentiated SH-5Y cells treated with supernatant from HIV infected U-937; GSM1045805 and GSM1045806 are normal differential SH-5Y cells selected as the negative control group. Differentiated SH-5Y cells are neuronal cell line cells which are widely used instead of primary neurons. Considering the difficulty of getting the clinical tissue samples of HAND patients (maybe impossible to obtain samples from early patients), we used DEGs between these cells reflecting the mRNAs/genes changing in neurons of HAND patients.

After the miRNA-mRNA network involved in HAND was constructed, we assigned the key miRNAs and mRNAs depending on the number of their neighbors/connections. This choice was somewhat arbitrary. Hausser and Zavolan reviewed that a miRNA binds to hundreds of sites across the transcriptome, so target mRNAs/genes are relatively excessive compared with miRNAs [51]. If we want to focus on the genes which are regulated by miRNAs, these mRNAs/genes regulated by many miRNAs simultaneously should be paid prior close attention. On the hand, these miRNAs, which regulate more mRNAs/genes, have a greater possibility to affect a pathway to exercise their biological functions. For instance, mir-7-3p, which is in our Table 1, was also found to regulate three target mRNAs/genes in the enriched KEGG pathway in Figure 4.

Finally, based on our constructed miRNA-mRNA network, we found a KEGG pathway–axon guidance pathway (hsa04360) through PPI network exploration and KEGG pathway enrichment. Some studies support our findings. A parallel genome-wide mRNA and miRNA profiling of the frontal cortex of HIV patients with and without HIV-associated dementia showed the role of axon guidance and downstream pathways in HIV-mediated neurodegeneration, and miR-137, 153, and 218 targeted neurodegeneration-related pathways [26]. Our study focused on the same axon guidance pathway, but different miRNAs and targets were provided. CXCL12, which was one of three genes in the pathway, and its receptor CXCR4, which is also the coreceptor of HIV, participate in the pathogenesis of CNS disorders such as HIV-associated encephalopathy, brain tumor, stroke, and multiple sclerosis [52]. Furthermore, one study of exosomal miRNAs associated with neuropsychological performance in individuals with HIV infection on antiretroviral therapy has shown that a high proportion of differentiating exosomal miRNAs target the axon guidance KEGG pathway, and the plasma neuronal exosomes can serve as biomarkers of cognitive impairment in HIV infection [28,53]. The axon guidance pathway and genes or miRNAs participating the pathway should be given enough attention in HAND research.

In summary, based on published data and publicly available databases, we constructed an integrated miRNA-mRNA network involved in HAND. On the basis of the miRNA-mRNA network, some key miRNAs and mRNAs/genes (listed in Table 1) were highlighted to some investigators who are interested in HIV-related neuronal damage. Furthermore, the axon guidance pathway, three genes of our network involved in this pathway, and those miRNAs which regulate them in our miRNA-mRNA network should also be explored by researchers. All of our findings provide new perspectives to the understanding of miRNAs and mRNAs underlying the HAND process, and experimental verification is needed to validate our predictions.

## 4. Materials and Methods

### 4.1. GEO Data Processing

The dataset GSE44265 provided by Sawaya BE. et al. (https://www.ncbi.nlm.nih.gov/geo/query/acc.cgi?acc=GSE44265 (accessed on 1 December 2021)) [44] was downloaded from the GEO database. In the dataset, GSM1045807 and GSM1045808 are differentiated SH-5Y cells treated with supernatant from HIV (clade B) infected U-937; GSM1045805 and GSM1045806 are normal differential SH-5Y cells as the negative control group. All samples were hybridized on the Affymetrix Human Gene 1.1 ST Array (Affymetrix, Santa Clara, CA, USA). All of the raw data were processed using *affy* and related R packages with Robust Multi-array Average approach for background normalization as per the package instruction [54]. Differentially expressed genes (DEGs) were selected according to changing folds more than 2 and a *p* value less than 0.05. The DEGs were divided into the upregulated DEG group and the downregulated DEG group.

### 4.2. MiRNA Selection and miRNA-Target Gene Prediction

A total of 101 upregulated miRNAs and 91 downregulated miRNAs were selected according to 12 published papers about HIV neural damage. To predict the miRNA-target genes of the corresponding miRNA, miRtarbase database (http://mirtarbase.mbc.nctu.edu.tw (accessed on 10 December 2021)) [46], miRDB database (http://mirdb.org/ (accessed on 10 December 2021)) [47], and Targetscan database (www.targetscan.org (accessed on 10 December 2021)) [48], three public databases, were applied to search the target genes. The union of all target genes from three databases was selected as the candidate dataset of miRNA-target genes of the corresponding miRNA.

### 4.3. Construction of miRNA-mRNA Regulatory Network

In order to construct the miRNA-mRNA regulatory network, upregulated miRNAs and their miRNA-target genes were intersected with genes of the downregulated DEG group; downregulated miRNAs and their miRNA-target genes were intersected with genes of the upregulated DEG group. Finally, 1458 upregulated-miRNA-and-downregulated-mRNA pairs and 1981 downregulated-miRNA-and-upregulated-mRNA pairs were selected to construct the miRNA-mRNA regulatory network involved HAND by Cytoscape software (version 3.7.1) [49]. 

### 4.4. Identification of Key miRNAs and mRNAs/Genes in miRNA-mRNA Regulatory Network

The important nodes (key miRNAs and genes) of the miRNA-mRNA regulatory network were predicted and explored by Cytoscape “cytoHubba” plugin [55]. In upregulated-miRNA-downregulated-mRNA network, the top 10 ranked net degrees upregulated miRNAs and the top 10 ranked net degrees downregulated mRNAs/genes were identified as key upregulated miRNAs and key downregulated mRNAs. Similarly, 10 key downregulated miRNAs and 10 key upregulated mRNAs were identified in the downregulated-miRNA-upregulated-mRNA network.

### 4.5. PPI Network Construction

To predict the biofunctions and interactions of the genes in the miRNA-mRNA network, the Cytoscape “ClueGo” plugin was applied [56,57]. The maximum sub PPI network which contained the largest number of the net genes was selected.

### 4.6. KEGG Analysis and the Pathway Visualization

KEGG analysis was performed using the KEGG pathway database (https://www.genome.jp/kegg/pathway.html (accessed on 20 January 2022)) [58]. To show the key KEGG pathway, an R/Bioconductor package named Pathview was applied to visualize the concerned KEGG pathway [50].

### 4.7. Venn Diagram

In order to show the miRNAs against three genes in the key KEGG pathway, an online Venn diagram tool Venny2.1 (https://bioinfogp.cnb.csic.es/tools/venny/ (accessed on 20 January 2022)) was applied. 

## 5. Conclusions


1The integrated miRNA-mRNA regulatory network involved in HAND was constructed.2Based on the node degree analysis, 20 potential key miRNAs and 20 potential key genes were provided to HAND investigators.3The axon guidance pathway (hsa04360), three genes (NTNG1, EFNB2, and CXCL12) and 17 miRNAs were highlighted participating in HAND progression.


## Figures and Tables

**Figure 1 pathogens-11-00407-f001:**
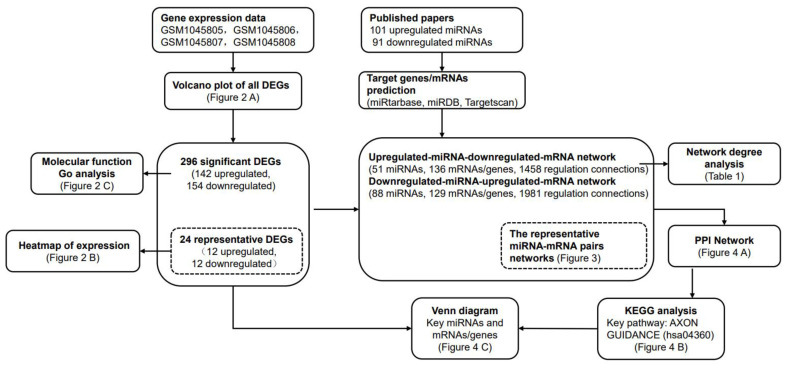
Flowchart of the study. An overview of the research strategy was shown, and the main study results were indicated in advance.

**Figure 2 pathogens-11-00407-f002:**
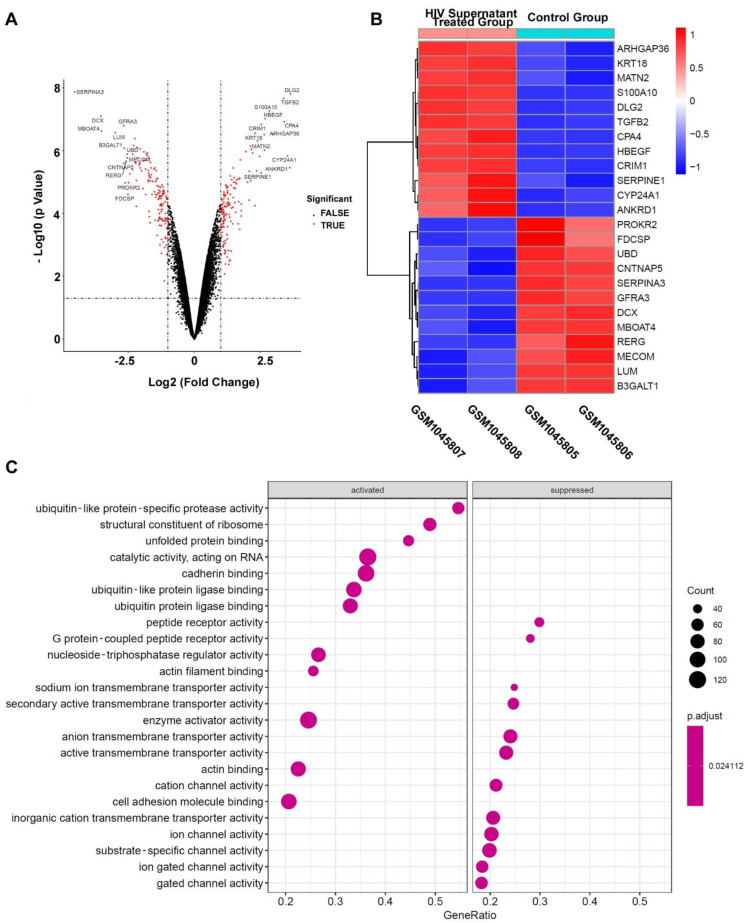
Analysis of DEGs associated with HAND. (**A**) Volcano plot of all DEGs. (**B**) Heatmap of representative DEGs. (**C**) Molecular function Go analysis of all 296 significant DEGs which were separated into the activated group and the suppressed group.

**Figure 3 pathogens-11-00407-f003:**
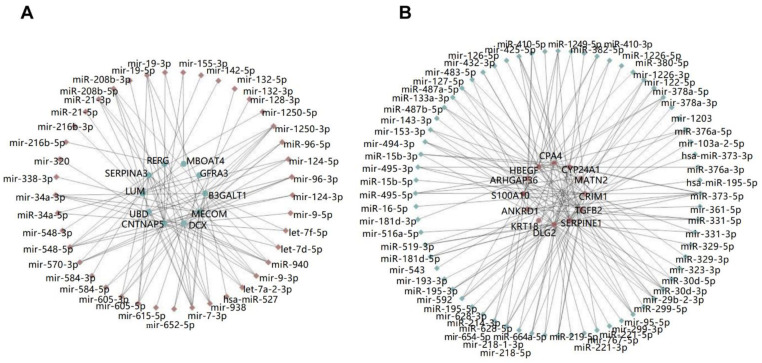
The representative miRNA-mRNA pairs networks. (**A**) The representative upregulated-miRNA-downregulated-mRNA network. (**B**) The representative downregulated-miRNA-upregulated-mRNA network.

**Figure 4 pathogens-11-00407-f004:**
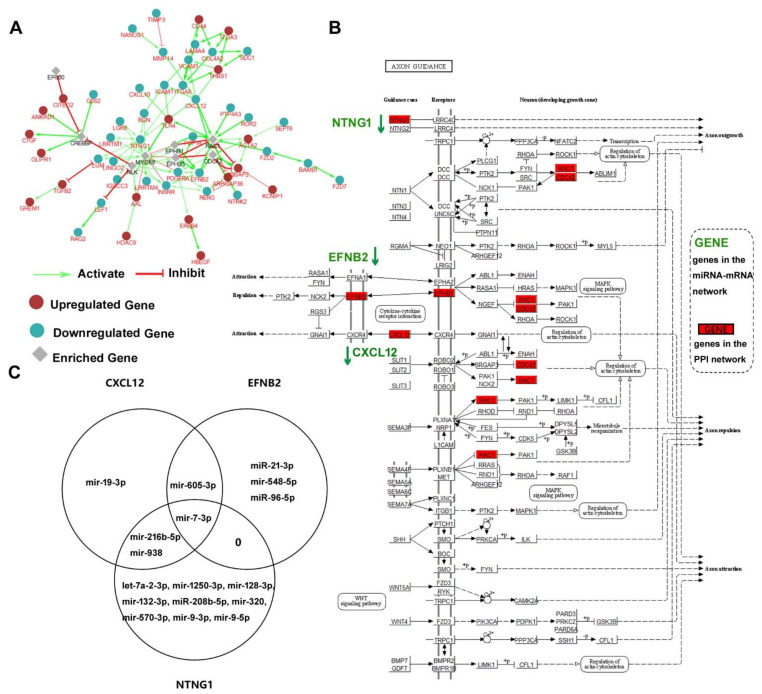
Functional exploration of the miRNA-mRNA network. (**A**) The PPI network of the mRNAs/genes in the miRNA-mRNA network. (**B**) Visualization of the axon guidance (hsa04360) KEGG pathway. Three genes of the miRNA-mRNA network were shown in green font. Seven genes also in the PPI network were indicated in red box. (**C**) Venn diagram showing miRNAs which regulate the three genes participating in the axon guidance pathway.

**Table 1 pathogens-11-00407-t001:** Top 10 up/down miRNA/mRNA in the miRNA-mRNA network base on degree analysis.

Upregulated miRNAs	Downregulated miRNAs	Upregulated mRNAs/Genes	Downregulated mRNAs/Genes
mir-570-3p	mir-95-5p	*LPP*	*CNNM2*
mir-7-3p	miR-373-5p	*STX7*	*FUT9*
mir-1250-3p	mir-126-5p	*DCLK1*	*SLC8A1*
miR-940	miR-30d-3p	*CCDC80*	*CLSTN2*
mir-548-5p	miR-195-3p	*ERBB4*	*IGSF3*
miR-208b-5p	miR-329-5p	*CADM2*	*NEBL*
mir-9-3p	mir-29b-2-5p	*PARVA*	*RAB27B*
mir-19-5p	miR-519-3p	*NRXN3*	*CACNA1C*
mir-548-3p	mir-103a-2-5p	*PLA2G16*	*DCX*
mir-124-5p	miR-380-3p	*DNAJB4*	*BAMBI*

## Data Availability

Not applicable.

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
