# Peer review of "Integrated Analysis of the miRNA-mRNA Regulatory Network Involved in HIV-Associated Neurocognitive Disorder"

_pathogens, 2022, doi:10.3390/pathogens11040407_

Round 1

Reviewer 1 Report

General comments:
The manuscript explore a really interesting topic but an understudied area. This is a well written paper. This study is attractive because the authors provide a evaluation about the roles of microRNAs in HIV-associated neurocognitive disorder (HAND). Furthermore, an integrated miRNA-mRNA network involved in HAND is contructed. This study with bioinformatics analyses provides new perspectives to the knowledge of miRNAs’ roles in the process of HAND. An improvement concerning this topic could serve as potential therapeutic targets. The work is well within the scope of the journal but it has some concerns. 

Specific comments:
-I think that its necessary include a schematic overview of the strategy used to derive the miRNA-mRNA interaction network (Figure 1) because this work is very interesting but maybe a bit difficult for readers to follow.

-There are different references of the list that are publications of at least 10 years ago (2000, 2001, 2003, 2004, 2007, 2008, 2009, 2010, 2011, 2012, 2013). I suggest, if this is possible, the citations need update, particularly in 2020 and 2021.

-I think that it would be included, a specific section of concluding remarks would be useful to readers.

Author Response

-I think that its necessary include a schematic overview of the strategy used to derive the miRNA-mRNA interaction network (Figure 1) because this work is very interesting but maybe a bit difficult for readers to follow.

Thank you for your advice. A flowchart of our study was presented as figure1. The flowchart descripted the whole research strategy, and indicated the locations of the main results in advance. We also added a new paragraph in the Result Section to descript our research strategy briefly. As we provided a new figure 1, the old figures 1, 2, 3 were marked as figure 2, 3, 4 successively. The manuscript was also modified accordingly.

-There are different references of the list that are publications of at least 10 years ago (2000, 2001, 2003, 2004, 2007, 2008, 2009, 2010, 2011, 2012, 2013). I suggest, if this is possible, the citations need update, particularly in 2020 and 2021.

Thank you for your good suggestion. Five more recent references (2020-2022) were added in our manuscript. Please check the details in the manuscript.

-I think that it would be included, a specific section of concluding remarks would be useful to readers.

Thank you for your comments. We added a specific section (5. Conclusions) to highlight our findings to readers. Please check the details in the manuscript.

Reviewer 2 Report

  1. In the results section on page 2, lines 88-91, it is stated that ‘…GSM1045805, GSM... were reanalyzed by R software’. Since this is the beginning of the results section, it is important to clearly state the identity of these samples and what they will be reanalyzed for.
  2. Figure 1 legend sounds like the authors treated SH-5Y cells with supernatant from HIV treated macrophages but in fact, they analyzed data from the GEO database. Please elaborate on the figure legend to clearly state sources of samples and what was done.
  3. What strain of HIV was used to treat the SH-5Y cells? This is important for readers who would want to make use of this data.
  4. Discussion of the results is limited but could be expanded by discussing some of the predicted functions of the miRNA-mRNA networks.
  5. In Figure 2 the miRNA names are not visible. Please increase font size.
  6. In Figure 3 legend, it will be helpful to add a key to the colors used in 3B.
  7. There are several language issues and typos that need attention.

Author Response

  1. In the results section on page 2, lines 88-91, it is stated that ‘…GSM1045805, GSM... were reanalyzed by R software’. Since this is the beginning of the results section, it is important to clearly state the identity of these samples and what they will be reanalyzed for.

Thank you for your advice. The information of the samples was provided as “Briefly described, GSM1045807 and GSM1045808 are differentiated SH-5Y cells treated with supernatant from HIV infected U-937; GSM1045805 and GSM1045806 are normal differential SH-5Y cells as the negative control group.”. Please check the manuscript.

  1. Figure 1 legend sounds like the authors treated SH-5Y cells with supernatant from HIV treated macrophages but in fact, they analyzed data from the GEO database. Please elaborate on the figure legend to clearly state sources of samples and what was done.

Thank you for the suggestion. We rewrote the figure2 (old figure 1) legend as “Analysis of DEGs associated with HAND”, and the detailed information about the samples and data processing was provided in “Results” and “Materials and Methods” Sections. Please check the manuscript.

  1. What strain of HIV was used to treat the SH-5Y cells? This is important for readers who would want to make use of this data.

Thank you for your advice. We check the information of GSE44265 and the citation of it. It can be confirmed that it was HIV-1 clade B, but the special strain is unknown. We added the information “HIV-1 clade B” in the Materials and Methods Section.

  1. Discussion of the results is limited but could be expanded by discussing some of the predicted functions of the miRNA-mRNA networks.

Thank you for your advice. Some of the predicted functions of the networks were added in the “Discussion” section. We also added a specific section (5. Conclusions) to highlight our findings to readers. Please check the manuscript.

  1. In Figure 2 the miRNA names are not visible. Please increase font size.

Thank you for your advice. The figure was replaced with the new figure which showed miRNA names clearly.

  1. In Figure 3 legend, it will be helpful to add a key to the colors used in 3B.

Thank you for your advice. We added the description content in Figure4 (old Figure 3) legend, and the figure was modified accordingly. Please check them in the manuscript.

  1. There are several language issues and typos that need attention.

Thank you very much for your advice. We have tried our best to improve the quality of English of our manuscript. Please check our detailed change in the manuscript.